# Pharmacist and Student Knowledge and Perceptions of Herbal Supplements and Natural Products

**DOI:** 10.3390/pharmacy11030096

**Published:** 2023-06-08

**Authors:** Jacey M. Stayduhar, Jordan R. Covvey, James B. Schreiber, Paula A. Witt-Enderby

**Affiliations:** 1Division of Pharmaceutical, Administrative, and Social Sciences, School of Pharmacy, Duquesne University, 600 Forbes Ave, Pittsburgh, PA 15282, USAcovveyj@duq.edu (J.R.C.); 2School of Nursing, Duquesne University, Pittsburgh, PA 15282, USA

**Keywords:** herbal supplements, natural medicine, holistic health, integrative health, pharmacists’ perceptions, pharmacy student perceptions

## Abstract

We aimed to collect parallel perspectives from pharmacists and pharmacy students on their use, knowledge, attitudes, and perceptions about herbal supplements/natural products. Two cross-sectional descriptive survey questionnaires—one focusing on pharmacists and the other focusing on pharmacy students—were administered from March to June 2021 via Qualtrics. The surveys were sent out to preceptor pharmacists and pharmacy students currently enrolled at a single U.S. school of pharmacy. The questionnaires were composed of five main sections, including (1) demographics; (2) attitudes/perceptions; (3) educational experience; (4) resource availability; and (5) objective knowledge of herbal supplements/natural products. Data analysis primarily utilized descriptive statistics with relevant comparisons across domains. A total of 73 pharmacists and 92 pharmacy students participated, with response rates of 8.8% and 19.3%, respectively. A total of 59.2% of pharmacists and 50% of pharmacy students stated they personally used herbal supplements/natural products. Most respondents (>95% for both groups) considered vitamins/minerals safe, although a lower percentage agreed on this for herbal supplements/natural products (60% and 79.3% for pharmacists and pharmacy students, respectively). Patient inquiries in the pharmacy setting were most seen for vitamin D, zinc, cannabidiol, and omega-3. A total of 34.2% of pharmacists reported having training in herbal supplements/natural products as a required part of their Pharm.D. training, and 89.1% of pharmacy students desired to learn more. The median score on the objective knowledge quiz was 50% for pharmacists and 45% for pharmacy students. Ultimately, herbal supplements/natural products are recognized by pharmacists/pharmacy students as a consistent and embedded part of pharmacy practice, although there is a need to enhance knowledge and skills in this area.

## 1. Introduction

Herbal product market sales in the United States (US) have demonstrated sustained increases for 12 consecutive years [1]. In 2020, US retail sales of herbal supplements totaled over an estimated USD 11.2 billion [1]. During the COVID-19 pandemic, herbal product market sales in the US increased by 17.3%, with a focus on immune health, inflammation, anxiety, sleep, and depression [1,2]. Data from the National Health Interview Survey (NHIS) have similarly demonstrated increases in the use of complementary (i.e., non-conventional interventions used with conventional interventions) and integrative (i.e., coordinated conventional and complementary interventions) approaches for managing health [3]. Specifically, the NHIS reported that approximately 59 million Americans spend USD 30.2 billion annually on natural products (dietary supplements other than vitamins and minerals) and mind/body practices such as yoga, meditation, and chiropractic services [4].

Even though the National Association of Boards of Pharmacy (NABP) has established guidelines on natural product competency in the North American Pharmacist Licensure Examination (NAPLEX) [5], criteria for integrative healthcare have not been established by the Accreditation Council for Pharmacy Education (ACPE). The National Center for Integrative Primary Healthcare (NCIPH) facilitated the development of a core set of interprofessional competencies and curriculum in integrative health care also supported by the ACPE standards 2 (Essentials for Practice and Care); 3 (Approach to Practice and Care); and 11 (Interprofessional Education), all which have key elements of complementary and integrative health as it relates to establishing competencies, patient advocacy, and interprofessional team education. Because herbal supplements/natural products are commonly sold in community pharmacies, consumers expect pharmacists to be knowledgeable and able to offer advice, especially given that in 2017–2018, more than half of US adults reported using a supplement [2,6]. Complementary and integrative health approaches, which are commonly used within treatment for chronic diseases (e.g., depression, cancer, chronic kidney disease, and diabetes) [7,8,9,10,11], can result in toxicities and harm if not properly understood and managed. As such, knowledge of these therapies for pharmacists is critical. While more than three-quarters of pharmacy curricula offer elective courses in complementary and alternative medicine (CAM) [12], there is further support for incorporation into the required curriculum with an emphasis on evidence-based practice [5,13,14,15,16,17].

Courses focusing on complementary approaches in pharmacy programs can effectively improve pharmacy students’ knowledge about complementary approaches. For example, in a description of one required course offered to second-year students, scores showed improvement in their knowledge of natural products; their ability to evaluate scientific literature and identify drug–food and drug–herb interactions; guide self-medication in a community setting; determine the correct products to stock and select for patients; and provide effective counseling, including advice on potential drug–herb contraindications [18]. In another course offered to third-year students, there was a positive change in pharmacy students’ perceptions of the value of various complementary medicines as well as in their willingness to recommend them, providing students with the required knowledge to make patient-centered recommendations for the use of complementary medicines in a professional pharmacy practice setting [19].

Recent surveys among pharmacists have revealed feelings of inadequate preparedness regarding this topic [20,21]. In one study with community pharmacists, only 34% reported having confidence in their ability to effectively counsel patients on herbal supplements, half (50%) reported never/rarely asking patients about herbal supplement use, and less than a quarter (~19%) reported that they always discussed herb–drug interactions [22]. Other recent studies have identified a lack of confidence on the ability to counsel, educate, and advise on herbal supplements/natural products, as well as lacking resources in the area [22,23,24].

Among pharmacy students, other studies have found agreement that knowledge was important, but their actual knowledge on the subject matter was limited [25,26]. These studies have also identified positive attitudes regarding the integration of CAM in conventional medicine and training within the Pharm.D. curriculum [25,26,27]. In the work setting, pharmacy students reported that they received questions about herbal supplements on a weekly-to-monthly basis, yet reported feeling discomfort handling questions, and perceived pharmacists they worked with to feel the same [28]. In one course evaluation, pharmacy students’ confidence in responding to patient questions, recommending specific herbal supplements/natural products, and capability of retrieving resources regarding these agents was low at baseline [29]. After completion of a course in this area, confidence improved [29], demonstrating the ability of education to address these gaps.

Previous studies [5,15,17,21,22,26,28,29,30] conducted on herbal supplements/natural products have primarily targeted either distinct samples of pharmacy students or pharmacists or focused on only one domain of assessment (perceptions, attitudes, competency, education, knowledge, or use). The present study sought to comprehensively integrate all relevant domains into a single assessment, identifying the necessary education to prepare future and practicing pharmacists to respond to their patients’ needs and keep pace with the rising trends of herbal supplement/natural product use. Accordingly, the objective of this study was to collect parallel perspectives from pharmacists and pharmacy students on their use, knowledge, attitudes, and perceptions about herbal supplements/natural products. From these data, the aim was to identify any gaps in knowledge and reasons for such gaps, so to design curricula to better respond to patients’ needs.

## 2. Methods

### 2.1. Study Design

Two cross-sectional studies using descriptive survey questionnaires (one focusing on pharmacists and one on pharmacy students) were approved by the university institutional review board (protocols 2022/01/09; 2022/01/10). The common aims of the surveys were to assess: (1) knowledge, perceptions, and interactions with herbal supplements/natural products; (2) confidence in abilities related to herbal supplement/natural product patient education/counseling; and (3) identification of educational needs for herbal supplements/natural products. The questionnaires were investigator-developed and based on a thorough literature review of the area [1,22,28], as well as the practice-based expertise of the corresponding author. The questionnaire was pre-tested with a small group of pharmacy students prior to survey administration, incorporating feedback on clarity and comprehensiveness into the final tool.

### 2.2. Participant Characteristics

The surveys were constructed using Qualtrics (Provo, UT, USA) and distributed via email from a single School of Pharmacy experiential education office from March to June 2022. The sample for the pharmacist survey consisted of actively practicing licensed pharmacists currently serving as preceptors associated with the School. The sample for the pharmacy student survey included students currently enrolled in the professional phases of the PharmD program at the School. The offer for voluntary survey participation was delivered alongside electronic consent. The survey offer was emailed to prospective participants a total of two times, including the initial offer and a one-time reminder.

### 2.3. Surveys

The questionnaires were composed of five main sections (Appendix A and Appendix B), including (1) demographics; (2) attitudes/perceptions about herbal supplements/natural products; (3) education on herbal supplements/natural products; (4) resources on herbal supplements/natural products; and (5) objective knowledge of herbal supplements/natural products. Closed-ended demographic items queried age, gender, practice setting details, years practicing as a pharmacist (or year of the professional program), and personal use of herbal supplements/natural products and mind/body practices, as relevant to the sample. A set of eleven and eight perception/education items were assessed for pharmacists and pharmacy students, respectively, using a four-point scale ranging from ‘*Strongly agree*’ to ‘*Strongly disagree*.’ Items queried participants regarding their perceptions of efficacy/safety of herbal supplements/natural products, the importance of herbal supplements/natural product knowledge and consultation, and confidence in their own knowledge/ability regarding herbal supplements/natural products. To provide context to these perceptions, participants were also provided a list of 22 different herbal supplements/natural products and asked to estimate the frequency with which they have interacted with patients regarding them (‘*Daily*’, ‘*Weekly*’, ‘*Monthly*’, ‘*Yearly*’ or ‘*Patients have not asked me*’); the list of products was based on market data and top-selling products over the last two years [1,2]. The final section utilized a 10-item objective knowledge assessment about herbal supplements/natural products [22], which was adapted with modification under a Creative Commons Attribution-Non Commercial License (CC BY-NC).

### 2.4. Data Analysis

Data analysis utilized descriptive statistics to describe responses to survey items. Scaled perception items were continuously represented with median and interquartile ranges (IQR), as well as categorically collapsed responses (‘*Strongly agree/agree*’ and ‘*Strongly disagree/disagree*’). Frequency of interactions with patients regarding herbal supplements/natural products was also categorically represented with collapsed categories (‘*Daily/weekly*’, ‘*Monthly/yearly*’, and ‘*Never’*). The objective knowledge assessment was scored according to each correct item as well as an overall mean (and standard deviation [SD]) score. Chi-square tests were utilized for comparisons of common perception items and patient interactions about herbal supplements/natural products between pharmacists and pharmacy students. Mann–Whitney U tests were utilized to compare the median scores on the knowledge assessment between the two groups. Results for items regarding perceptions and knowledge were stratified based on whether participants reported personal use of herbal supplements/natural products or mind/body practices (either one or the other, or both); this analysis also used Mann–Whitney U tests.

## 3. Results

### 3.1. General Results

Offers to participate in the surveys were emailed to a total of 830 pharmacists and 476 pharmacy students, and corresponding response rates were 8.8% and 19.3%, respectively. Basic demographics and characteristics for the 73 pharmacists and 92 pharmacy student respondents are shown in Table 1. Among the pharmacists, 40.9% were 24–34 years of age, and 32.4% were 35–44 years of age. Most participants were women working in the hospital pharmacy setting. Among the pharmacy students, 88% were 21–25 years of age, with the majority being women spread across all years in the professional curriculum, and primarily working for community chain pharmacies. Use of herbal supplements/natural products ranged between 50–60% and was similar among pharmacists and pharmacy students. Nearly two-thirds of pharmacists reported personal use of mind/body practices, compared to just under half of pharmacy students (*p* = 0.032).

### 3.2. Perceptions of Herbal Supplements/Natural Products

Perceptions of pharmacists and pharmacy students toward aspects of herbal supplements/natural products were variable (Table 2). Overall, the majority of both groups considered vitamins/minerals and herbal supplements/natural products safe, with benefits in prevention of disease and symptomatic treatment. Perceptions about the utility of herbal supplements/natural products to cure disease were less confident. Across items common to both surveys, pharmacy students were more likely to consider vitamins/minerals safe for use (90.4% for pharmacists vs. 98.9% for pharmacy students; *p* = 0.01), herbal supplements and natural products safe for use (60.0% vs. 79.3%; *p* = 0.007), and that there are benefits to the use of herbal supplements and natural products to cure disease (22.4% vs. 48.3%; *p* < 0.001). When pharmacists were asked about their perception about the prevalence of herbal use during their years in practice, a total of 43 (58.9%) pharmacists reported that they perceived increased prevalence of patients using herbal supplements/natural products compared to 28 (38.4%) indicating no change in prevalence, and 2 (2.7%) indicating lower prevalence.

When perceptions were stratified according to the participant’s report of personal use of either herbal supplements/natural products or mind/body practices, there were some differences identified (Table 2). Pharmacists who reported personal use of herbal supplements/natural products were more likely to identify these products as beneficial for preventing disease (*p* = 0.001) and treating the symptoms of some diseases (*p* < 0.001). They were also more confident in their ability to counsel patients on the side effects associated with herbal products (*p* = 0.04). For pharmacists reporting the personal use of mind/body techniques, they were more likely to endorse perceptions regarding the benefits of herbal supplements/natural products in curing diseases (*p* = 0.033). Students reporting personal use of herbal supplements/natural products had more positive perceptions for five of the six statements assessed (all *p* < 0.05), except for the perception of the safety of vitamins/minerals, where no difference was seen. The stratification of personal use of mind/body practices did not generate any differences in perceptions.

### 3.3. Interactions with Patients about Herbal Supplements/Natural Products

Table 3 describes the frequency with which participants reported patients inquiring about each herbal supplement/natural product. For both groups, vitamin D, zinc, cannabidiol, and omega-3 were the most common products that patients inquired about, with approximately one-third to one-half of participants identifying such encounters on either a daily or weekly basis. On the opposite side of the spectrum, inquiries about mushrooms, beet root, fenugreek, wheat/barley grass, valerian, and ashwagandha were very infrequent or not inquired about at all. When pharmacists were asked about encounters with patients for drug–herb interactions and side effects, 14 (16.5%) stated that discussion occurs when a patient purchases a product, 64 (75.3%) during a patient inquiry about a product, and 7 (8.2%) reported never having discussions.

### 3.4. Education on Herbal Supplements/Natural Products

Among pharmacists, 25 (34.2%) reported that training in herbal supplements/natural products was a required part of their PharmD training, compared to 25 (34.2%) reporting elective coursework and 23 (31.5%) reporting no inclusion at all. Among pharmacists that completed a pharmacy practice residency (*n* = 33), only three (9.1%) reported the inclusion of herbal supplement/natural product training. A total of 38 (52.1%) reported completion of continuing education credits related to herbal supplements/natural products.

For students, desire to learn more about herbal supplements/natural products was endorsed by 82 (89.1%) participants. A total of 30 (36.6%) students identified that required coursework in herbal supplements/natural products would be preferable, compared to 52 (63.4%) that favored elective coursework. When asked how they perceived the feelings of their professors toward herbal supplements/natural products, 9 (10.0%) rated the feelings as positive, 53 (58.9%) as neutral, and 28 (31.1%) as negative.

### 3.5. Knowledge about Herbal Supplements/Natural Products

For the 10-question herbal knowledge quiz, the median score was 5 for pharmacists and 4.5 for pharmacy students (*p* = 0.00168). The overall median scores for each group were unchanged when stratified according to personal use of herbal supplements/natural products and mind/body practices. Performance on individual items is shown in Table 4. The questions that posed the most difficulty for participants included those regarding the use of ginseng for colds (9–23% correct among the two participant groups), the safety of echinacea among patients with autoimmune disease (16–38% correct), and the safety of chamomile for patients with diabetes (21% correct). Higher-performing items included awareness of the need to prove safety/efficacy for herbal products before marketing (84–92% correct) and the regulation of herbal products by the Food and Drug Administration (FDA) under the federal Food, Drug, and Cosmetic Act (87–90% correct).

When pharmacists were asked whether evidence-based resources regarding herbal supplements/natural products were readily available at work, 42 (57.5%) agreed, 26 (35.6%) disagreed, and 5 (6.8%) did not know. Regarding whether they knew where to locate reliable information, 49 (67.1%) strongly agreed/agreed while 24 (32.9%) strongly disagreed/disagreed. When pharmacy students were asked whether they had access to evidence-based resources within their school, 11 (12.0%) were in agreement, 11 (12.0%) disagreed, and 70 (76.1%) did not know.

## 4. Discussion

This study conducted parallel assessments of pharmacists’ and pharmacy students’ perceptions, interactions, and knowledge of herbal supplements/natural products to identify gaps in education and practice. The data reveal that participants generally had low confidence regarding counseling patients on their herbal supplement/natural products. Although pharmacists and pharmacy students had differing perceptions regarding herbal supplements’/natural products’ ability to cure disease, both cohorts did perceive they had utility in preventing disease and symptomatic treatment. Through the objective knowledge assessment, it was also identified that significant knowledge gaps for herbal supplements/natural products were present for both groups. Although pharmacists achieved a higher overall score, both cohorts failed to correctly answer half the quiz questions, which was slightly lower than scores published from a previously published analysis [22]. The questions that posed the most difficulty for participants included those related to the efficacy/safety of specific supplements for treating acute and chronic conditions.

One of the more important findings from this study was how personal use of herbal supplements/natural products or mind/body practices impacted respondent perceptions. In our study, two-thirds of the pharmacists surveyed used herbal supplement/natural products or mind/body practices, which was greater than the pharmacy students surveyed, reporting just under one-third [30]. These data contrast with other studies, where over half of respondents used some form of complementary approach [30,31]. Alongside the objective knowledge limitations identified by the quiz, such a combination of perceptions may be problematic for patients if pharmacists lean more heavily on personal experience over objective knowledge within counseling, making the counseling ineffective if not harmful. That being said, personal use of complementary approaches by pharmacists and pharmacy students is thought to contribute to their readiness and ability to make recommendations to patients [15,32]. As such, these findings suggest that enhancing objective knowledge on herbal supplements/natural products or mind/body practices should be a goal for pharmacists. This is critical considering that most pharmacists surveyed indicated an increase in inquiries about herbal supplements/natural medicines. A previous study identified that among working pharmacy students, 79% were asked questions about herbal supplements/natural products on an at least monthly basis [28]. Accordingly, a sound knowledge foundation combined with personal experience may produce the best outcomes for patients.

In exploring participant educational experiences, it was clear that experiences were variable across the board. Approximately two-thirds of practicing pharmacists reported some level of education available (either required or elective) within their Pharm.D. training, and approximately half had completed continuing education on the topic as practicing pharmacists. Pharmacy students indicated a clear desire for enhanced education in this area, yet with approximately one-third reporting that they perceived their professors did not feel positively about herbal supplements/natural products. These results connect directly with our results on limited confidence and lower objective knowledge. A previous survey by Camiel et al. found that 83% of pharmacy students were not comfortable processing herbal/dietary supplement questions and 65% believed that pharmacists were also not comfortable [28]. However, both groups in our survey agreed that it was important to be knowledgeable about herbal supplements/natural products, indicating that there is significant motivation to close these gaps. Close to two-thirds of pharmacy students stated a desire for this content as an elective course, similar to other studies [19]. The knowledge deficit observed for both practicing pharmacists and pharmacy students argues for more rigorous and standardized training in this area, especially in the areas of counseling on the use of herbal supplements/natural products as it relates to drug/herb interactions and patient risk factors; proper documentation of herbal supplement/natural product use by patients; identifying appropriate evidence-based resources when determining herbal supplement/natural product safety and drug interactions; and critical analysis of the research on herbal supplement/natural product safety and efficacy.

A course devoted specifically to herbal supplements/natural products, especially herb–drug interactions, would allow for content to be taught in a coherent and focused manner, rather than a sprinkling of this content throughout the curricula. Currently, pharmacy curricula teach herbal supplements/natural products primarily with didactic content. However, this content may become diluted throughout the curricula and these concepts are not retained. As stated by Lee et al., reinforcement of these principles should be given to pharmacy students early and in later professional years through evidence- and application-based courses to expand pharmacy students’ skills and information base throughout their education [6]. This would also enable competencies in herbal and natural medicine to be assessed in a rigorous manner discussed in the Introduction. Our study suggested that faculty development training on herbal supplements/natural medicines could complement course offerings to offset misconceptions, which our survey demonstrated to be a strong contributing factor to students’ knowledge and preparedness.

The pharmacist is the healthcare professional who is best poised to meet changes in practice and interest for herbal supplements/natural products through the provision of evidence-based counseling. Beyond education, state boards of pharmacy need to better specify herbal supplement/natural product resources in practice settings as they are reported to be inconsistent [30], and pharmacies should provide sufficient number, quality and access to information resources while providing additional education for their practicing pharmacists and interns. In a previous study, when asked about the availability of resources on this topic, only 7% of respondents reported that their pharmacy had adequate resources, while 46% reported having no access and 35% reported having no experience with resources on herbal supplements/natural products [28]. The findings from the present study indicated better access, but still suboptimal overall. In the long term, more governmental investment in funding robust and evidence-based research on complementary and integrative health will help offset the dearth of research reported by systematic reviews in this area, which demonstrate that less than 8% of CAM therapies are evidence-based [33,34]. 

## 5. Limitations

First, the study was sampled at a single school of pharmacy, limiting the generalizability of the results as well as the demographic diversity of respondents. While the study had a small sample, it was representative of the expected demographics of the source population at this single institution. Second, although a broad sample was ultimately captured, the survey response rendered a smaller sample size overall, meaning that not all experiences may have been captured and represented. Third, although all professional years (PY1-PY4) were surveyed, content leaned towards being more heavily taught in the later years (PY2-4) of education at this institution, which could have impacted responses. Fourth, statements for attitudes/perceptions assessed an individuals’ opinions on herbal supplements/natural products overall; it should be recognized that individuals may have different opinions specific to individual products, the nuance of which was unable to be captured through these items. Fifth, the survey developed for this analysis was not validated, although there were no other validated instruments available the authors felt fit the needs of the study. Sixth, many of the items included in this survey relied on self-reported data. Finally, the herbal knowledge quiz, while used in previous work, contained only ten objective knowledge items on different focused topics; accordingly, it may not be fully representative of an individual’s knowledge on the entire landscape of herbal supplements/natural products.

## 6. Conclusions

Overall, this study demonstrated a need to improve education/training in herbal supplements/natural products. This is based on our results revealing (1) objective knowledge deficits regarding herbal products, (2) variable inclusion of herbal supplements/natural products education within the Pharm.D. curriculum, and (3) a desire to learn more coupled with increased patient interest in such products. Enhanced educational/training efforts offers the opportunity for pharmacists to feel more confident and knowledgeable in patient interactions related to herbal supplements/natural products across practice settings.

## Figures and Tables

**Table 1 pharmacy-11-00096-t001:** Participant demographics and characteristics.

Variable, *n* (%)	Pharmacists (*n* = 73)	Pharmacy Students (*n* = 92)
Gender		
Male	23 (32.4)	18 (19.6)
Female	50 (70.4)	74 (80.4)
Other	0 (0.0)	0 (0.0)
Primary practice setting *		
Hospital	27 (38.0)	19 (20.7)
Chain	11 (15.5)	42 (45.7)
Ambulatory care	10 (14.1)	-
Independent	8 (11.3)	14 (15.2)
Academia	5 (7.0)	-
Long-term care	5 (7.0)	3 (3.3)
Managed care	4 (5.6)	-
Specialty	3 (4.2)	-
Government	1 (1.4)	-
Other	5 (7.0)	9 (9.8)
Personal use of herbal supplements/natural products	42 (59.2)	46 (50.0)
Personal use of mind/body practices	44 (62.0)	40 (43.5)
Year in school		
PY1	-	20 (21.7)
PY2	-	19 (20.7)
PY3	-	21 (22.8)
PY4	-	32 (34.8)

N = number; PY = professional year; * may not add up to 100% as participants could select more than one response.

**Table 2 pharmacy-11-00096-t002:** Participant perceptions regarding herbal supplements and natural products.

Perception Statements	Pharmacists				Pharmacy Students			
	*n*	Median (IQR) *	Strongly Agree/Agree (%)	Strongly Disagree/Disagree (%)	*n*	Median (IQR) *	Strongly Agree/Agree (%)	Strongly Disagree/Disagree (%)
In general, I consider the use of vitamins and minerals safe.	73	3 (3-3)	90.4	9.6	92	3 (3-3)	98.9	1.1
In general, I consider the use of herbal supplements and natural products safe.	70	3 (2-3)	60.0	40.0	92	3 (3-3) ^a^	79.3	20.7
I believe there is a benefit for utilizing herbal supplements and natural products to prevent disease.	71	3 (2-3) ^ac^	71.8	28.2	91	3 (2-3) ^a^	73.6	26.4
I believe there is a benefit in using herbal supplements and natural products to treat symptoms of some diseases.	72	3 (3-3) ^ac^	80.6	19.4	90	3 (3-3) ^a^	83.3	16.7
I believe there is a benefit in using herbal supplements and natural products to cure diseases.	67	2 (1-2) ^b^	22.4	77.6	89	2 (2-3) ^a^	48.3	51.7
As a pharmacist, it is important to be knowledgeable about herbal supplements and natural products.	73	4 (3-4)	98.6	1.4	-	-	-	-
I am confident in my ability to counsel patients on herb–drug interactions.	71	3 (2-3)	54.9	45.1	-	-	-	-
I am confident in my ability to counsel patients on herbal side effects.	71	2 (2-3) ^a^	46.5	53.5	-	-	-	-
I need to gain additional knowledge to properly counsel patients on herbal supplements and natural products.	72	3 (3-4)	84.7	15.3	-	-	-	-
I am confident in my ability to recommend or discourage use of herbal supplements and natural products based on patient-specific factors.	71	3 (2-3)	60.6	39.4	-	-	-	-
I believe it is important to consult a health care provider (MD, PharmD, etc.) before utilizing herbal supplements or natural products.	72	3 (3-4)	91.7	8.3	-	-	-	-
I consider myself knowledgeable about herbal supplements and natural products.	-	-	-	-	92	2 (2-3) ^a^	28.3	71.7
I consider my professors knowledgeable about herbal supplements and natural products.	-	-	-	-	88	3 (2.75-3)	75.0	25.0
I think it is important to be knowledgeable about herbal supplements and natural products as a future pharmacist.	-	-	-	-	92	4 (3-4)	96.7	3.3

IQR = interquartile range; *n* = number; * Responses rated on a 4-point scale with 4 = Strongly agree, 3 = Agree, 2 = Disagree, and 1 = Strongly disagree; ^a^ significant difference according to whether the respondent reported personal use of herbal supplements/natural products; ^b^ significant difference according to whether the respondent reported personal use of mind/body practices; ^c^ significant difference according to whether the respondent reported personal use of both herbal supplements/natural products AND mind/body practices.

**Table 3 pharmacy-11-00096-t003:** Participant interactions with patients about herbal supplements and natural products.

Herbal Supplement/Natural Product	Pharmacists (%)			Pharmacy Students (%)		
	Daily/Weekly	Monthly/Yearly	Never	Daily/Weekly	Monthly/Yearly	Never
Echinacea	2.8	45.1	52.1	1.1	18.5	80.4
Turmeric	9.9	59.2	31.0	3.3	38.0	58.7
Elderberry	11.3	42.3	46.5	5.4	41.3	53.3
Ginger	2.8	46.5	50.7	6.5	47.8	45.7
Garlic	2.8	52.1	45.1	6.5	37.0	56.5
Fenugreek	0.0	23.9	76.1	1.1	6.6	92.3
Saw palmetto	0.0	50.7	49.3	3.3	26.4	70.3
Flax seed/oil	2.8	40.8	56.3	1.1	25.0	73.9
Aloe vera	1.4	42.3	56.3	9.8	46.7	43.5
Wheat/barley grass	0.0	21.1	78.9	1.1	9.8	89.1
Valerian	0.0	36.6	63.4	2.2	13.0	84.8
Milk thistle	0.0	43.7	56.3	2.2	20.7	77.2
Ginkgo	1.4	59.2	39.4	2.2	17.4	80.4
Ginseng	1.4	52.1	46.5	2.2	18.5	79.3
Beet root	0.0	19.7	80.3	1.1	7.6	91.3
Ashwagandha	7.0	25.4	67.6	3.3	14.1	82.6
Apple cider vinegar	9.9	43.7	46.5	7.6	45.7	46.7
CBD	35.2	46.5	18.3	29.7	45.1	25.3
Vitamin D	50.7	36.6	12.7	58.7	19.6	21.7
Zinc	32.9	51.4	15.7	47.8	22.8	29.3
Omega-3	36.6	43.7	19.7	38.0	25.0	37.0
Mushrooms	1.4	17.1	81.4	2.2	3.3	94.6

Sample responses ranged *n* = 70–71 for pharmacist group and *n* = 91–92 for pharmacy student group, as not all participants answered each item.

**Table 4 pharmacy-11-00096-t004:** Breakdown of results on the herbal knowledge quiz.

	Pharmacists (%)			Pharmacy Students (%)		
	True	False	Unsure	True	False	Unsure
Herbal products are regulated by the FDA under the federal Food, Drug, and Cosmetic act.	8.5	87.3 *	4.2	5.4	90.2 *	4.4
Safety and efficacy of herbal products must be proven before they are marketed.	5.6	91.6 *	2.8	6.5	84.8 *	8.7
St. John’s wort is commonly used for mild–moderate depression.	84.5 *	8.5	7.0	66.3 *	8.7	25
Echinacea is a safe treatment for patients with autoimmune disease.	7.0	38.0 *	54.9	9.8	16.3 *	73.9
Garlic can increase risk of bleeding.	78.9 *	2.8	18.3	65.2 *	4.4	30.4
Ginkgo biloba is commonly used for dementia.	60.6 *	9.9	29.6	38.0 *	5.4	56.5
Ginseng is commonly used for the treatment/prevention of the common cold.	33.8	22.5 *	43.7	34.8	9.8 *	55.4
Saw palmetto is contraindicated in men with BPH.	25.4	47.9 *	26.8	30.4	14.1 *	55.4
Chamomile is safe for diabetic patients.	21.1 *	4.2	74.7	21.7 *	8.7	69.6
Ginger is contraindicated in pregnancy.	5.6	43.7 *	50.7	12	28.3 *	59.8

* Denotes correct answer for each item.

## Data Availability

The data presented in this study are available on request from the corresponding author.

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
