# Peer review of "Pharmacist and Student Knowledge and Perceptions of Herbal Supplements and Natural Products"

_pharmacy, 2023, doi:10.3390/pharmacy11030096_

Round 1

Reviewer 1 Report

The study provides an interesting and comprehensive overview on the role of natural product based medicines for practicing and prospective pharmacists. Methods have been described in detail and limitations, such as the low number of responders, have been discussed sufficiently. Apart from two minor suggestions for correction, I would recommend the manuscript for publication.

Line 277: "...recommended patients...": "to" seems to be missing.

I would not recommend to present the most important contents in bold writing, but I guess that's to be decided by the editors.

Reviewer 2 Report

MDPI Manuscript Review – 5/22/23

Pharmacist and Student Knowledge and Perceptions about Herbal Supplements

Overall comments:

The manuscript was not proofread before submission. There are numerous typographical errors. In addition, the manuscript is not written in one voice. Revise to improve flow.

Abstract: Needs revision for clarity and grammatical/typographical errors. Consider adding in data and statistics from results section.

Lines 11&12 – not a complete sentence

Lines 15-17 – check punctuation

Line 20 – “Most respondents” include percentage

Line 23 – “most common” – should be written as “most commonly”

Lines 24 and 25 – consider using percentage instead of “one-half or “two-thirds” for clarity

Line 24 – word “interaction” in the pharmacy world connotes drug interactions, consider using another word such as “encounters”

Lines 25-27 – Sentence starting “Approximately two-thirds…was a required...” words are missing “which was a required elective?” “…although scoring on an objective knowledge…?” Is the word quiz missing? Second half of the sentence needs further clarification.

Not sure why lines 20-27 are bolded.

Introduction:

In first paragraph of the introduction, consider making parenthesis on lines 39-41 and 43-44 part of the sentence (restructure the sentences).

Lines 45-48 – “Within pharmacy education…” sentence structure is unclear

Line 45 - NABP is the National Association of Boards of Pharmacy

Line 48 – ACPE is the Accreditation Council for Pharmacy Education

Line 50 – omit “in the past 30 days”

Lines 62 and 63 – consider using percentage rather than half, less than half

Line 64 – word is missing “…lack of confidence ? the ability…”

Lines 68 – there is an extra word in the sentence, omit “regarding” or reword

Lines 69-71 – change to past tense

Lines 74 – 75 – split into 2 sentences

Line 76 - “significantly” please provide statistics

Lines 77-80 – provide references for these previous studies

Line 85 – change “this data” to “these data”

Lines 83-87 – clarify study objectives

Methods

Overall, this section is clearly written.

Results

Lines 190, 192, and 194 – include percentages/data with p-values

Table 3 – Combining/collapsing data from monthly/yearly does not give the reader a clear idea of how frequently patients ask about supplements as these are not similar time frames. Consider separating the data.

Lines 206 and 207 – Using “interactions” and “drug-herb interactions” in the same sentence is confusing

Discussion

This section should elaborate on the study results and how it relates to your literature review in the introduction. Paragraphs starting on lines 306, through the end of the discussion section (Line 353) should be included in the introduction/literature search section and referred to in the discussion section.

Limitations

An additional limitation could be that students in all professional years were included in the study. The investigators did not specify if P1 or P2 students learn about herbal/natural products in the curriculum and where this would take place. Some students, such as those in the first professional year (P1) might not have learned about herbal/natural products, and therefore, would not have the same knowledge as students in more advanced professional years.

Conclusion

The conclusion does not fully sum up the study results and discussion. Include additional information regarding pharmacist and patient confidence in herbal products

Line 371 –In the U.S., herbal supplements are labeled as not being intended to diagnose, treat, cure, or prevent disease. Perhaps reword to say … government-sponsored research to determine safety and efficacy of herbal products.”

Survey instrument:

Section on Attitudes also includes questions (9-11) on demographics. The questions on attitudes should be listed separately.

Line 625 – Student survey, Question 13 asks about years of practice as a pharmacist. These are pharmacy students or student pharmacists.

Please see above comments 

Reviewer 3 Report

In this manuscript, the authors conducted a survey to collect parallel perspectives from pharmacists and pharmacy students regarding their use, knowledge, attitudes, and perceptions about herbal supplements and natural products. Two separate cross-sectional descriptive survey questionnaires were administered to actively practicing pharmacists serving as preceptors and pharmacy students enrolled at a single US school of pharmacy. 

Here are some comments:

1.       The study was conducted at a single US school of pharmacy, which may not adequately represent the diverse perspectives and experiences nationwide. The findings should be interpreted within the context of this limitation and not generalized to the entire pharmacist and pharmacy student population.

2.       The study's sample size is relatively small due to the low response rates of 8.8% for pharmacists and 19.3% for pharmacy students (73 participating pharmacists and 92 pharmacy students). This limited sample size raises concerns about the generalizability of the findings and may undermine the validity of the conclusions drawn. The significant difference in response rates between the two groups introduces potential bias and could affect the representativeness of the pharmacist sample. The study should address this issue and discuss possible strategies for improving response rates in future research.

3.       As the study relies on self-reported data, there is a risk of self-report bias, which could impact the reliability of the information obtained. The authors should acknowledge this limitation and consider additional measures to validate the self-reported responses, such as cross-referencing with other data sources or conducting follow-up interviews.

4.       Is question 45 in Appendix A an open-ended survey question? The authors should consider summarizing the meaningful insights derived from the open-ended responses if applicable.

5.       Additional data analysis could enhance the study. Exploring differences among various age groups, years of working experience, different practice settings, or pharmacy student class cohorts would provide valuable results that may be helpful in identifying potential trends or patterns.

6.       Detailed responses to the herbal quizzes should be provided to serve as a reference for other educational institutions or researchers interested in evaluating knowledge levels in this area.

7.       In Table 2, it would be beneficial to present identical questionnaires for pharmacy students and pharmacists side by side. This would make it easier for readers to compare and check the responses between the two groups.
